# A Review of Conductive Hydrogel Used in Flexible Strain Sensor

**DOI:** 10.3390/ma13183947

**Published:** 2020-09-07

**Authors:** Li Tang, Shaoji Wu, Jie Qu, Liang Gong, Jianxin Tang

**Affiliations:** 1Hunan Key Laboratory of Biomedical Nanomaterials and Devices, College of Life Science and Chemistry, Hunan University of Technology, Zhuzhou 412007, China; tangli_352@163.com (L.T.); wushaoji321@163.com (S.W.); qujie9876@163.com (J.Q.); 2State Key Laboratory of Chemo/Bio-Sensing and Chemometrics, College of Chemistry and Chemical Engineering, Hunan University, Changsha 410082, China

**Keywords:** flexible strain sensor, hydrogel, conductivity, stretchability

## Abstract

Hydrogels, as classic soft materials, are important materials for tissue engineering and biosensing with unique properties, such as good biocompatibility, high stretchability, strong adhesion, excellent self-healing, and self-recovery. Conductive hydrogels possess the additional property of conductivity, which endows them with advanced applications in actuating devices, biomedicine, and sensing. In this review, we provide an overview of the recent development of conductive hydrogels in the field of strain sensors, with particular focus on the types of conductive fillers, including ionic conductors, conducting nanomaterials, and conductive polymers. The synthetic methods of such conductive hydrogel materials and their physical and chemical properties are highlighted. At last, challenges and future perspectives of conductive hydrogels applied in flexible strain sensors are discussed.

## 1. Introduction

With the development of science and technology, smart wearable devices have been widely used in human health monitoring, human movement detection, human–machine interfaces and soft robotics. The emergence and development of smart wearable devices brought the challenge to the corresponding devices to meet higher demands for mechanical deformation matching including bending, folding, twisting, and stretching etc. Compared to traditional devices fabricated by metals or semiconductors, which only can sense small deformations less than 5% strain and are accompanied by inherent rigidity making them uncomfortable to wear, flexible strain sensors (FSSs) have proved to be a promising candidate for smart wearable devices due to their flexibility, light weight and biocompatibility (Figure 1). Especially, FSSs could be designed to meet large-range sensing requirements that convert mechanical deformation into a measurable signal (e.g., electronic signal). Conventional flexible sensors are made by coating or filling with conductive, materials including carbon-based nanomaterials, metal-based nanomaterials, and conductive polymers, onto elastomers (e.g., Ecoflex, polydimethylsiloxane (PDMS)) and rubbers (natural rubber and thermoplastic elastomers (TPEs)) [1,2,3,4]. Although they indeed provide a simple approach for fabrication and exhibit good electrical conductivity, the poor stretching capacity (2–300%) is still insufficient to meet the high strain requirements of FSSs [5]. Soft and sensitive flexible sensors with high stretching capacity are urgently needed for the next generation of wearable and portable electronic product markets.

As a soft and wet material, hydrogels with unique properties of swelling behavior, flexibility, good biocompatibility and porosity, have demonstrated their versatility in many research and industry fields, including biomedical engineering [10,11], sensor and actuator [12,13], agriculture and wastewater treatment [14,15]. However, the fragile and brittle properties of hydrogels constitute an obstacle to extend their further applications in wearable devices and tissue engineering. To solve these issues, much effort has been devoted for mechanical strengthening of hydrogel, such as double network and interpenetrating network strategies. Recent advances in toughening hydrogel showed that double network (DN) hydrogels [16,17,18], nanocomposite (NC) hydrogels [19,20,21], and double crosslinked hydrogels [22,23] possess strong mechanical properties which can withstand extreme deformation. Moreover, these hydrogels exhibited some interesting characters, such as adhesive property, self-healing property and self-recovery property [24,25,26,27]. When toughness of hydrogel is combined with conductive material, those properties make composite hydrogels to be an outstanding candidate for FSSs. Compared with traditional devices with poor deformation capacity, smart wearable devices based on such FSSs with toughened hydrogel show high stretchability, flexibility, high sensitivity, light weight, fast response, and stability [28,29,30]. More importantly, high stretchability, lower stretching-releasing hysteresis, good self-recovery and durability properties of composite hydrogel endowed the sensor a stable electrical signal change in wide linear range (0–1000%) without significant damage. Meanwhile, we illustrated the publication numbers regarding to “hydrogel”, “hydrogel” and “flexible strain sensor” from 2010 to 2019 via Web of Science. As shown in Figure 2, it clearly shows that hydrogels have received increasing attention. Hydrogels based FSSs became a new direction to expand hydrogel applications in 2017, and continued to show explosive growth in 2019.

To provide practical guidelines, in this mini review, we summarized the recent advances of conductive hydrogel used in FSSs, and discussed their conductive mechanisms exhibited by the various gels filled with different conductive materials (for example, ionic conductors, carbon-based nanomaterials, metal-based nanomaterials etc.). Representative examples are selected here and their methods of synthesis, features and distinctions with future prospects are discussed. To be specific, the review was divided into five parts. In the first section, we give an introduction of flexible strain sensor and hydrogel. In the second, third, and fourth sections, conductive hydrogels are reviewed based on the types of conductive fillers, including ionic conductors, conducting nanomaterials, and conducting polymers. At last, we provide a conclusion and future prospect of conductive hydrogel used in FSSs, in order to present possibilities that could help shape the future developments of a truly wearable strain sensor.

## 2. Ionic Conductors

Directly doping soluble inorganic salts into hydrogel is a simple and straightforward approach to prepare conductive hydrogel. High water content and microporous structure of hydrogels are favorable for dispersion of the resolvable salts, which permit ionic conductivity for the hydrogels. According to the role of ions in conductivity hydrogel, the ionic hydrogel was classified as physical doping and chemical doping. Some inorganic ions just as free ions doping into the gel to improve the conductivity of hydrogel without formation of new bonding, were called physical doping agents. Another is chemical doping that ions act as coordination centers to crosslink the different polymer chains in hydrogel to form hydrogel network. According the role of ions in gel, the reported various ionic conductive hydrogel used as FSSs and their properties are summarized in Table 1. Gauge factor (GF) was used to define the sensitivity of conductive hydrogel, and it calculated by GF = (ΔC/C_0_)/ε or GF = (ΔR/R_0_)/ε for capacitive-type and resistance-type strain sensors, respectively. The GF of most ionic conductive hydrogel was located between 0.0125 and 6. These gels showed a high stretchability from 500% to 3120% over that of the traditional FSSs (2% to 300%). Also, these ionic conductive hydrogels performed the unique properties of hydrogel, such as adhesion, self-healing, self-recovery, and biocompatibility, etc.

### 2.1. Free Ion

The most effective way to endow hydrogels with good conductivity is directly dissolved strong electrolyte in the hydrogel precursor solution, which would provide abundant cation and anion as carriers. The conductivity and sensitivity of hydrogel depend the concentration of doped ions. Yang and Yuan reported an agar/PAAm DN hydrogel incorporated with LiCl for fabricating strain sensor (Figure 3a) [37]. As a conductive filler, the LiCl electrolyte was firstly dissolved in hydrogel precursor solution, and then the mixed solution formed the DN hydrogel with trapped LiCl via heating-cooling-polymerization method. The conductive hydrogel could light a luminous diode when applied it in a closed circuit. Also, it demonstrated an extremely broad strain window (0−1100%) with a high GF of 1.8 at 1100% strain. As illustrated in Table 1, directly doping salts in hydrogel for preparing strain sensors got rapid development. While simple to fabricate, there are still many problems to be solved in practical applications for such strain sensors. For example, in the application of wearable electronic devices, a high ion concentration is helpful to improve the conductivity of sensors, but it is not profit for maintaining the biocompatibility of sensors because high ion concentration is harmful to cells and tissues. To this end, Zhao and co-workers introduced a new strategy that encapsulate high conductivity salt-solution patterns in PEG matrix to fabricate conductive hydrogel (Figure 3b) [42]. The resultant salt/poly (ethylene glycol) aqueous two-phase systems (salt/PEG ATPS) hydrogel can not only detect bending, stretching, pressing, and other motion signals, but can also be used for biologically matched electronic interfaces due to its high biocompatibility.

### 2.2. Crosslinked Ion

During the fabrication process, some natural polymer like agar and gelatin, could form gels by physical heating-cooling method. Besides, other natural polymer needs to be crosslinked by ion (Fe^3+^, Ca^2+^, Li^+^) for forming hydrogel, such as alginate, κ-carrageenan, hyaluronic acid (HA). Benefiting from the abundant crosslinked ions, the hydrogels crosslinked by cation demonstrated an excellent conductivity and high performance in the strain sensor.

Alginates are linear anionic polysaccharide containing (1→4′)-linked β-d-mannuronic acid and α-l-guluronic acid, and it has been widely used in many hydrogels because of its high hydrophilic, strong adhesion and non-toxicity. For example, Sun and co-workers presented a polyacrylamide/alginate (PAAm/alginate) DN hydrogel with the second alginate network crosslinked by Ca^2+^ (Figure 4) [45]. The addition of Ca^2+^ not only dissipated part of energy by unzip of Ca^2+^ crosslinked alginate to improve the mechanical property of hydrogel, but also endowed the DN hydrogel with conductivity. In addition, the strain sensor performed extremely tensile strain (~1700%) and showed wide linear range (20%–800%), stable and reliable response signal (200 cycles) and fast response time (800 ms). Furthermore, the strain sensors were used to detect laughing and talking, and it also could distinguish between rapid deep breathing and regular breathing. Xia and co-workers prepared a dual physically cross-linked P(AAm-*co*-LMA)/alginate DN hydrogel based on the first P(AAm-*co*-LMA) network crosslinked by core-shell hybrid nanoparticles and the second alginate network cross-linked by Ca^2+^ [46]. When the strain sensor was stretched to 100%, the resistance signal change (ΔR/R_0_) of the hydrogel reached 200%. Benefiting from its ultrasensitive property, the strain sensor was applied to monitor tiny motions, such as breathing and speaking.

Despite having excellent mechanical and electrical properties, these ionic hydrogels, lacking self-healing, adhesion, and antifreeze properties, are not compatible in harsh mechanical conditions. To obtain these functionalities, some monomers with special property and strategies were employed in hydrogel.

κ-carrageenan is another linear sulfated polysaccharide which extracted from red algae. It not only can be crosslinked by cations, including Ca^2+^, K^+^, Li^+^, but also could form thermoreversible gel via heating-cooling process. The thermoreversible property of monomer is one of most efficient method to achieve self-healing property of hydrogel. Under high temperature, the broken hydrogel was turned to sol state, and after cooling it can self-healed and reformed to bulk hydrogel. For example, Liu and Li et al. fabricated a physical and chemical hybrid κ-carrageenan/PAAm DN hydrogel, in which the first κ-carrageenan network was crosslinked by K^+^ and the second PAAm network was crosslinked by chemical agents [47]. The strain sensors not only produce electrical signal to different levels of finger bending degree in real time, but also exhibited a great strain sensitivity with a GF of 0.63 at the strain of 1000%. Benefiting from the thermoreversible property of κ-carrageenan, the self-healing behavior of DN hydrogel was observed at 90 °C, and the self-healed hydrogel also showed an excellent conductivity (99.2%). Moreover, taking advantage of the high toughness and plasticity of hydrogels, they successfully used the warm κ-carrageenan/AAm pre-solution as an ink to print complex 3D patterns with prominent mechanical property. Wang and Qiu reported a strain sensor based on the hybrid agar/calcium alginate (CA)/polyacrylamide (PAAm) conductive hydrogels through 3D printing [48]. Firstly, mixture of agar/alginate/AAm was extruded as a continuous stripe and it can maintain its shape due to high viscosity. After UV photopolymerization and CaCl_2_ solution soaking, the strain sensor with GF of 3.83 and linear range between 0 and 200% was obtained.

Wu et al. reported antifreeze DN hydrogels and organohydrogels based on κ-carrageenan/PAAm system by introducing ethylene glycol (Eg) and glycerol (GI) into hydrogel, providing a solvent-replacement strategy to fabricate antifreeze strain sensor (Figure 5) [44]. The hydrogel showed good stability and repeatability from −18 to 25 °C with different strain levels, which can detect the 0.5%–50% strain signal even at −18 °C. This work provides a generic method to fabricate an antifreezing conductive hydrogel, and it solved the problem that traditional conducive hydrogel with weak stretchability and conductivity.

## 3. Conducting Nanomaterials

Although metal ions can provide outstanding ionic conductivity for hydrogel strain sensors, it will inevitably cause adverse electrochemical reactions at the contact interface, which is not propitious to the long-term stable operation of the sensor. Therefore, doping nanomaterials with high conductivity instead of metal ions into hydrogels have been recognized as one of the most common strategy to fabricate conductive hydrogel. According to the composition of chemical elements, those conductive nanomaterial fillers can be divided into carbon nanomaterials and metal-based nanomaterials [54,55,56,57,58].

### 3.1. Carbon-Based Nanomaterials

Carbon nanomaterials are natural or artificial materials composed of carbon with size ranging from 1 nm to 1 μm, including carbon nanotubes (CNTs), graphene, carbon fibers (CFs) and carbon black (CB). They have been widely used in biosensing, electro/photocatalysis and lithium-ion batteries, due to their special physical/chemical structure, excellent electrical conductivity, large specific surface area and easy functionalization [59,60,61,62]. Among the above carbon materials, CNTs and graphene have attracted more attention in biosensing, energy harvesting and drug delivery. Distinctive advantages of CNTs, such as high aspect ratio, excellent conductivity, and excellent stability in wet environments, have made them an ideal candidate for conductive hydrogel fillers. However, due to poor surface functional groups, it is difficult to dissolve and disperse in the solvent. To this end, researchers modified the surface of CNTs for fabricating conductive hydrogel, including carboxyl functionalization, amino functionalization or organic molecular modification. Gao group reported a carboxyl-functionalized multi-walled carbon nanotube (c-MWCNTs) crosslinked chemical-physical hybrid hydrogel [63]. Conductive c-MWCNTs crosslinked chitosan (CS) through dynamic ionic bonds to form physical crosslinked hydrogel. C-MWCNTs not only provided hydrogel with excellent, stable and repeatable resistance response signal (linear range 0–500%, 300 cycles) as the hydrogel stretched, but also endowed the hydrogel with self-healing and self-recovery property as its crosslinked by dynamic physical bonds. The results showed that the conductivity of resultant hydrogel increased with increasing concentration of c-MWCNTs, which could reach to ~ 1 S/m at 1.25 wt.% c-MWCNTs. Based on the strategy, nanomaterials could enhance the mechanical property and polydopamine could improve the adhesive property of the hydrogel. For example, Lu group incorporated polydopamine-decorated CNTs into p(AAc-*co*-AAm) covalent network using glycerol/water as the solvent of hydrogel (Figure 6) [64]. The hydrogel displayed an excellent conductivity and adhesion property under a harsh environment from −20 to 60 °C. Particularly, the maximum value of conductivity was 8.2 S/m and adhesion strength was 57 ± 5.2 kPa at 10 wt.% PDA-CNTs when glycerol volume contents was 50 vol%. Moreover, when the concentration of CNTs increased from 0 to 15 wt.%, the tensile stress of hydrogel may improve from 20 to 60 kPa. From the above reports, dopamine modification to CNTs could improve their dispersion and reduce the agglomeration of CNTs in hydrogel, which was helpful for enhancing the conductivity of hydrogel by increasing the content of CNTs.

Graphene is a type 2D carbon nanomaterial that composes of single layer of carbon atom with sp^2^ hybrid hydrocarbon framework. It is an excellent carbon-based conductive filler with high carrier mobility and large surface area. Lv and co-worker reported a new strategy to fabricate strain sensor based on graphene nanomaterial [65]. They coated graphene on the surface of hydrogel to form a continuous graphene film instead of mixing graphene with hydrogel pre-solution to form gel. The fabricated graphene/hydrogel composite material showed highly sensitive to strain, for example, the ΔR/R_0_ of the hydrogel was up to 60% when the hydrogel stretched at ~25%. Although the method can produce highly sensitive sensors, it only has a very small stretch sensing linear range (0–25%). Due to the continuous graphene film is easy to produce cracks and thus failed to maintain stable electric signals. While the method can fabricate highly sensitive sensor, the continuous graphene film is easy to produce cracks and thus failed to maintain stable electric signals. Nevertheless, if combine the conductive nanomaterial fillers with ionic conductor to obtain conductive hydrogel, the resulting hydrogels were not only high sensitive to deformation, but also showed various of other interesting properties, such as biocompatible, self-healing, and adhesion. Inspired by mussels, Turng’s group combined dopamine (DA), polyacrylic acid (PAAc), Fe^3+^, and reduced graphene oxide (rGO) to prepare hydrogel strain sensor with self-healing, biocompatible, and high ductility property [66]. In this system, the dynamic binding between the carboxylic acid group of PAAc and Fe^3+^ provided the sensor with good self-healing performance. While improving the dispersion of rGO nanomaterials in hydrogels, DA also endowed the hydrogel with fast self-healing property and strong adhesion ability via forming the dynamic physical cross-links between the sensors and substrate. Moreover, rGO provided the effective electric pathways which enabled hydrogel with high strain sensitivity and it could be used to detect multiple human motions.

### 3.2. Metal-Based Nanomaterials

Metal-based nanomaterials have both the superior electrical conductivity of bulk metals and properties of nanomaterials including surface plasma resonance (SPR), catalytic property, mechanical property and magnetic property. The outstanding properties make them have extensive potential application in the fields of light, electricity, catalyst, antibacterial, biosensing and biomedicine [67,68,69,70]. Metal-based nanomaterials were also excellent nano-fillers to prepare conductive hydrogel due to their good conductivity. Xu and co-worker synthesized an ultra-stretchable, highly sensitive, transparent and biocompatible capacitive strain sensor based on silver nanofibers ionic hydrogel (Figure 7a) [71]. Capacitive hydrogel strain sensor, whose capacitance decreased with the hydrogel stretched, has a similar of sensing mechanism with the abovementioned resistive types strain sensor. Benefit from the strong electrical conductivity of silver nanofibers, the nanocomposite hydrogel exhibited high sensitivity to the deformation. The maximum gauge factor (GF) could reach 165, which is much larger than that of the pure ionic conductive hydrogel (GF < 10). It is worth noting that the stable electrical signal change of sensors only obtained at small deformation (cycling stretch between 50% to 150%) and could not obtained effective electrical signal when the cyclic tensile deformation is great than 400%. Chen and co-worker reported another ultrasensitive conductive hydrogel consisting with silver nanowires, graphene oxide, calcium chloride and PVA (Figure 7b–c) [72]. In this system, silver nanowires network displayed high conductivity, while graphene oxide nanosheets endowed the system with rich functional group and Ca^2+^ acted as bridges to connect PVA and graphene oxide nanosheets. Under the optimized concentration, the GF of conductive hydrogel could reach ~3500 at cyclical ~50% strain. The sensitivity was enhanced by around four order of magnitudes compared to the pure ionic conductive hydrogel. The high sensitivity of strain sensor made it possible to accurately detect and monitor the small strain. For example, the sensors enabled precise detection of the wrist pulse, and it could clearly distinguish the diastolic wave, tidal wave, and percussion wave from pulse wave form.

## 4. Conducting Polymers

Polymer is macromolecule with molecular weight of up to several thousand or even hundreds of thousands Dalton. Some long-chain polymers mainly composed of carbon atoms and conjugated π-electron systems exhibiting excellent electrical conductivity, known as conductive polymers. Conductive polymers have several advantages than other conductors, including (1) more uniform dispersion than nanomaterials because it can be synthesized in situ, (2) easiness to adapt for deformation and difficult to cause dislocation because of its flexibility, and (3) higher conductivity property than ionic conductor. In recent years, a variety of conductive polymers with different conductivity properties have been reported, such as poly (3, 4-ethylenedioxythiophene): polystyrene sulfonate (PEDOT: PSS), polyaniline (PANi), polypyrrole, and polythiophene.

### 4.1. PEDOT: PSS

PEDOT is a π-conjugated conductive polymer synthesized in 1988 with a high conductivity, but the poor water solubility has seriously affected its application. Mixing PSS and PEDOT can perfectly solve this problem, but it also hinders the carrier transport and reduces the conductivity of PEDOT. Several methods have been used to improve the conductivity of PEDOT: PSS, for example, doping and compounding with carbon-based material or protonic acid. These approaches greatly promote the application of PEDOT in the fields of light-emitting diode, organic solar cell, organic thin film transistor and supercapacitor [73,74,75,76,77,78]. Naficy and co-workers prepared PPEGMA1100/PAA conductive DN hydrogel with different mechanical property, pH response property, electrical conductivity property, and swelling property by adjusting the polymerization steps of the PEDTO: PSS [79]. The result indicated that DN hydrogels with PEDTO: PSS polymerized by two times (DN-PEDOT(PSS)-II) had a better performance than that polymerized by one time (DN-PEDOT(PSS)-I) in electrical conductivity. With increased polymerization steps, the amount of PEDOT increased and even could reach to the maximum. Also, further increasing the polymerization steps did not lead a higher conductivity (DN-PEDOT(PSS)-I with conductivity of 3.7 × 10^−3^ S/cm, DN-PEDOT(PSS)-II with conductivity of 3.4 S/cm and DN-PEDOT(PSS)-III with conductivity of 4.3 S/cm). By measuring the resistance of the DN-PEDOT(PSS)-II hydrogel under compression strain, it can be found that a larger compression strain occupied lower resistance, indicating that compressed hydrogel will help to increase the conductivity of hydrogel. In addition, the feature of hydrogel fabricated by mixing PEDOT: PSS and functionalized-multiwalled carbon nanotubes (F-MWCTs) also have been investigated. Ye and co-worker reported chemically and physically crosslinked conductive CNT-PEDOT-PAM-PVA hydrogel, which could quickly and sensitively respond to the mechanical deformation [80]. Furthermore, they packed the conductive hydrogel into a pressure microsensor for detecting the pulse of the radial and carotid arteries via printing and injecting method.

### 4.2. PANi

Polyaniline is formed by end-to-end connection of aniline molecules, containing a reducing benzene unit and an oxidizing oxime unit. The eigenstate polyaniline is a poor conductive polymeric material; however, it could obtain excellent electrical conductivity through doping substance which usually is proton. Chen and co-worker designed a self-healing conductive hydrogel by introducing PANi doped proton into supramolecular hydrogels [81]. The conductivity of optimized hydrogel could improve to 13 S/m, and exhibited a stable GF ~3.4 in the linear strain range between 0 to 300%. Although it presented a high conductivity, the poor mechanical properties still exist. To solve this problem, Duan and co-worker reported a novel microsphere-structured hydrogel (MC-Gel) based on chitosan microsphere and PANi/phytic acid (Figure 8) [82]. The hydrogel exhibits resistance change responsiveness to both compression and stretching. Furthermore, a 5000 times loading–unloading test was employed with pressure changed from 0 to 1 kPa, and the hydrogel presents an excellent reproducibility of resistance variations and fatigue resistance property.

### 4.3. Polypyrrole

Polypyrrole is a conductive polymer which prepared by chemical oxidizing or electrochemical oxidizing a five-membered heterocyclic pyrrole. The repeated arrangement of the single-double bond alternating conjugated structures on the polypyrrole backbone provides a channel for the movement of the π electrons. Gu and co-worker reported macroporous conductive hydrogels (PC hydrogels) based on stiff polypyrrole and soft hydrogel using sugar particles as pore-foaming agent [83]. The PC hydrogel displayed excellent conductivity as the polypyrrole was uniform dispersed in hydrogel network and exhibited outstanding fatigue resistance due to the macroporous structure could effectively dissipated energy. However, the hydrophobic property of polypyrrole makes it difficult to combine with hydrogel. In order to overcome this deficiency, Lu and co-workers reported a strategy for enhancing the hydrophilicity of polypyrrole via polydopamine (PDA) modification [84]. The transparent and adhesive hydrogel strain sensor showed high conductivity with 12 S/m at the optimal conditions.

## 5. Conclusion and Future Prospect

In the past decades, much effort has been paid by researchers for the development of flexible conductive hydrogel strain sensors, and great progress is evidenced. In order to present possibilities that could help shape the future developments of a truly wearable strain sensor, advantages and disadvantages of conductive hydrogel according different conductive fillers were summarized in Table 2. As shown, the sensitivity or gauge factor of conductive hydrogel could be tuned from 0.125 to 7790 by choosing different conductive fillers. The hydrogel strain sensor exhibited extremely strong deformability and can be easily pulled up to over twenty times its original length, while the traditional flexible strain sensors can only be pulled several times over its original length. Also, parts of conductive hydrogels possess some special performances, including biocompatibility, anti-freezing, good biocompatibility, strong self-adhesive ability, self-healing, and fast self-recovery property, which will be more conducive to its application in a human–machine interface, health monitoring, and human motion (Figure 9).

However, challenges still remain, including:(1)A superior candidate flexible strain sensor should have the following characteristics: high stretchability, good biocompatibility and anti-freezing property, strong adhesive, fast self-healing, and self-recovery. However, most reported hydrogel cases hardly to occupy all of the above features at the same time;(2)It is very important and meaningful that the sensor can correctly distinguish irregular pulse and heart beats signals from complex environments for monitoring human health, but most hydrogel strain sensors were only sensitive to huge human motion, including different stretching, finger/leg bending, speaking, and in only a few cases was it used for monitoring tiny pulse and heart beats. So, it is still a big challenge to develop highly sensitive sensors.(3)Most researches were focused on the strain sensors performance studies rather than the packaging, integration, and practical application of hydrogel strain sensors.

## Figures and Tables

**Figure 1 materials-13-03947-f001:**
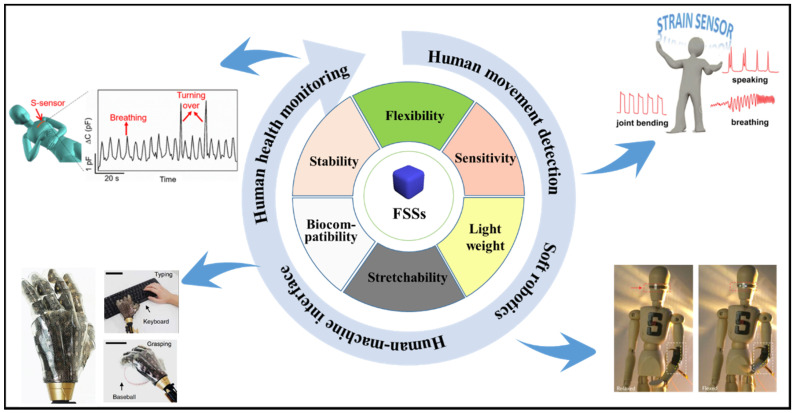
Summary of the properties and applications of flexible strain sensors [6,7,8,9].

**Figure 2 materials-13-03947-f002:**
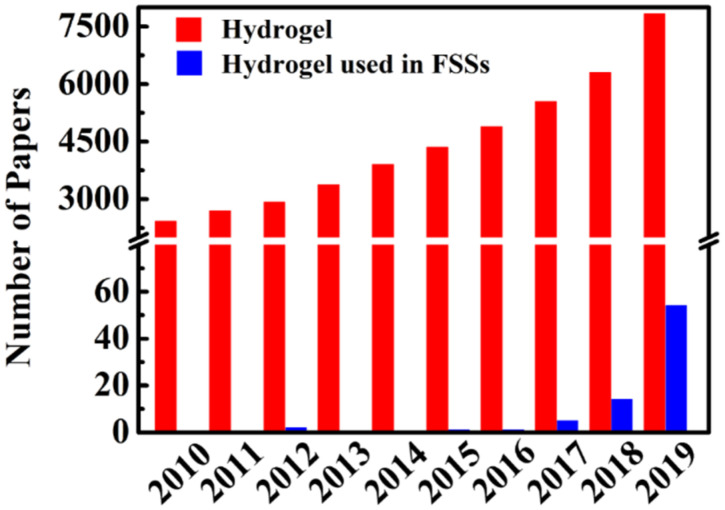
Number of published papers per year, by searching the topic “hydrogel”, “hydrogel” and “flexible strain sensor” from Web of Science in 05/2020.

**Figure 3 materials-13-03947-f003:**
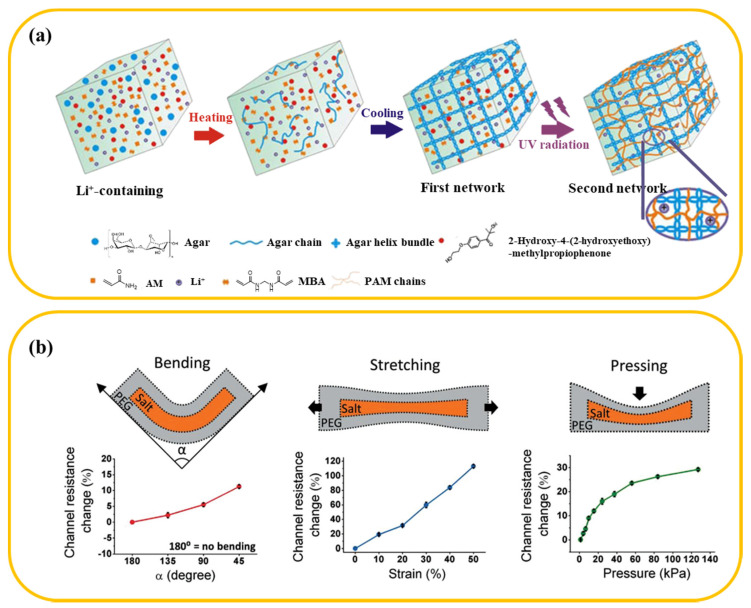
(**a**) Schematic of fabricating the Li^+^/agar/PAM ionic double-network hydrogels via one-pot method, (**b**) High-conductivity salt-solution patterns are stably encapsulated within PEG hydrogel matrices and its response to mechanical inputs, including bending, pressing, and stretching.

**Figure 4 materials-13-03947-f004:**
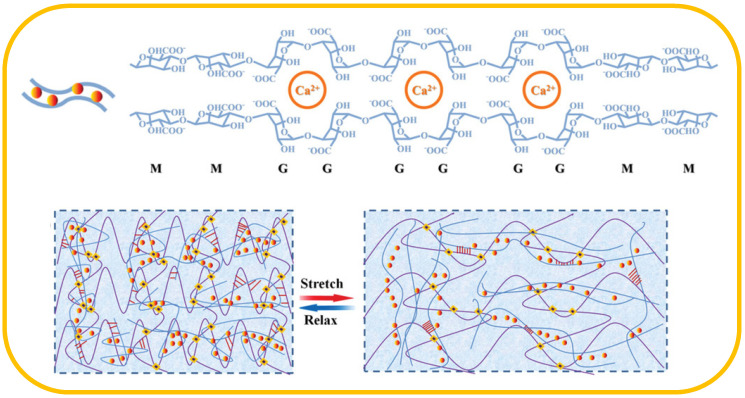
Structure of alginate crosslinked by Ca^2+^ and schematic diagram of the variation of the double network in hydrogel under stretching and releasing.

**Figure 5 materials-13-03947-f005:**
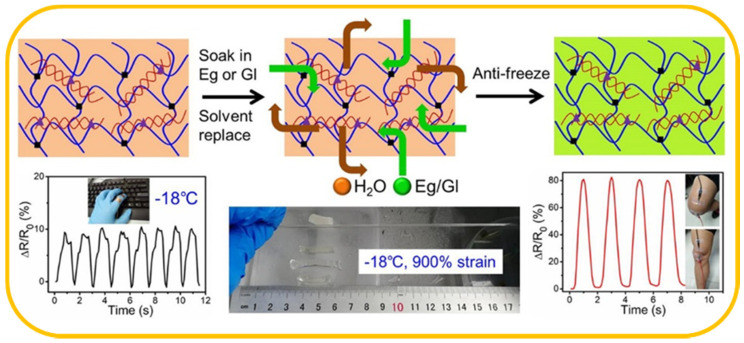
Schematic of a solvent-replacement strategy to fabricate antifreeze strain sensor and the sensitivity of sensor under different conditions.

**Figure 6 materials-13-03947-f006:**
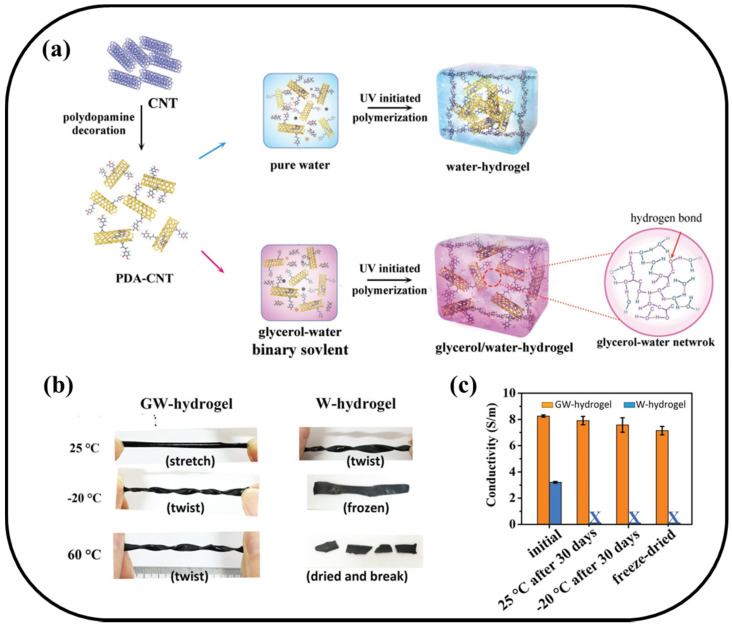
(**a**) Schematic of fabricating a conductive hydrogel with extreme temperature tolerance based on glycerol–water binary solvent. (**b**) The anti-freezing and anti-heating performance and (**c**) Conductivity of GW-hydrogel and W-hydrogel under different temperatures.

**Figure 7 materials-13-03947-f007:**
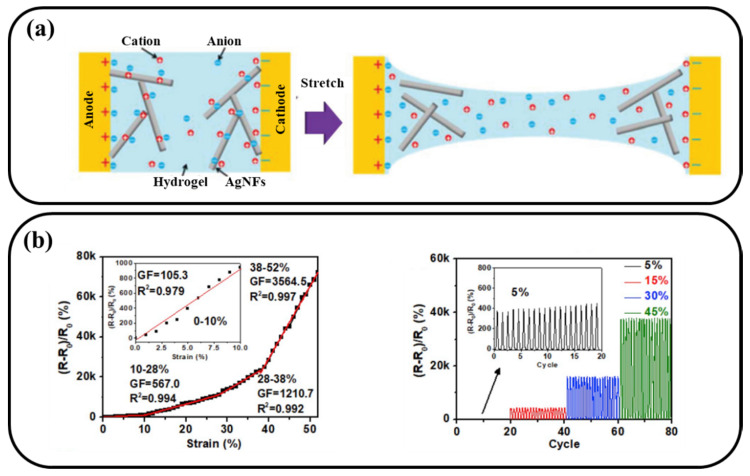
(**a**) Schematic of strain sensor based on brick-mortar structure, (**b**) Gauge factor and (**c**) Sensing behavior of strain sensor under different strain, respectively.

**Figure 8 materials-13-03947-f008:**
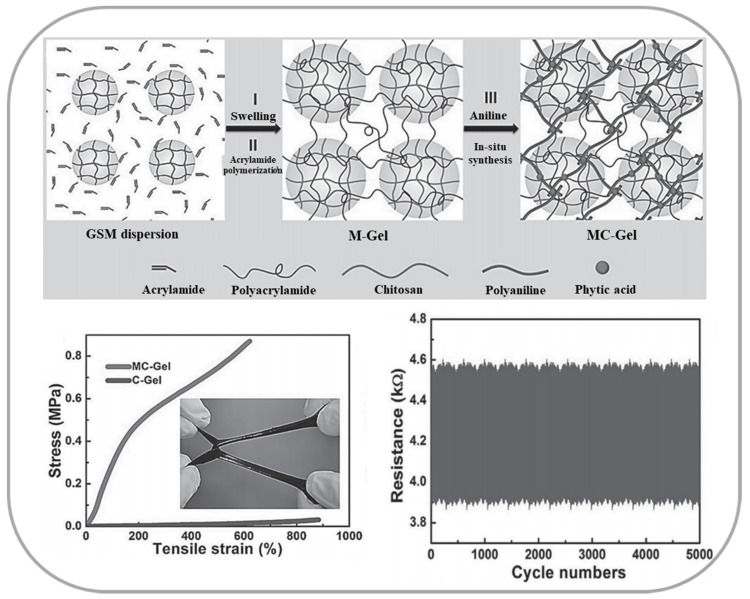
The schematic preparation process, mechanical property and resistance variations under 5000 cycles of the MC-Gel.

**Figure 9 materials-13-03947-f009:**
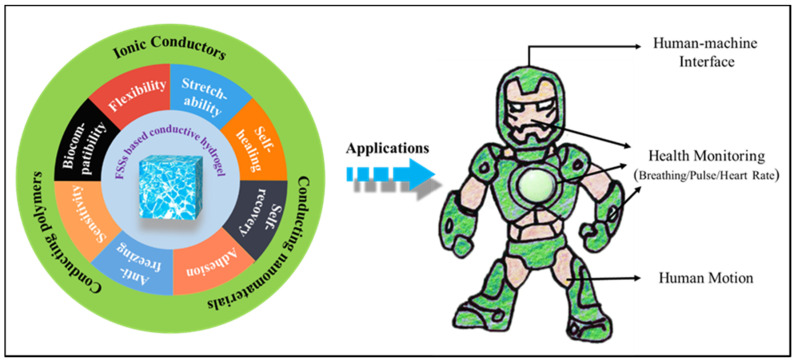
The properties and applications of FSSs based on conductive hydrogel.

**Table 1 materials-13-03947-t001:** Summary of the performance of reported various ionic conductive hydrogel used as flexible strain sensor.

Type	Hydrogel	Electrolyte	Properties	Ref.
**Free ion**	PAAm	KCl	GF = 0.125–0.693, stretchability (~1000%), self-recovery, adhesion, self-healing, linear range (0–1000%)	[31]
	PAAm	Sodium casein	Conductivity, stretchability (~2100%), adhesion (2200 N/m), linear range (0–300%)	[32]
	PEO/PAAm	LiCl	Conductivity (~8 S/m), stretchability (~880%), self-healing	[33]
	Cellulose	NaCl	GF = 0.297, antifreeze, linear range (0–230%)	[34]
	PAAm	LiCl	GF = 0.4, stretchability (~1465%), antifreeze, adhesion	[35]
	P(HMA-AAm)	NaCl	GF = 2.37, stretchability (~2160%), self-recovery, anti-fatigue, linear range (0–1000%)	[36]
	Agar/PAAm	LiCl	GF = 1.8, stretchability (~1650%), flexible electroluminescent, linear range (0–1100%)	[37]
	PVP/PVA	FeCl_3_	GF = 0.478, stretchability (~1160%), self-recovery, adhesion, self-healing, tensile stress (2.1 MPa)	[38]
	PVA/PAAc	H_2_SO_4_	Self-recover, stretchability (~2600%), toughness (18.7 MJ m^−3^), fracture stress (3.1 MPa), linear range (0–500%)	[39]
	Agar/PAAm	NaCl	GF = 2.1, stretchability (~1920%), linear range (0.5%–1600%)	[40]
	Poly(LysMA-*co*-AAm)	LiCl	Conductivity (0.0425–0.0736 S/cm), stretchability (~2422%), antifreeze, adhesion	[7]
	Adenine/thymine/P(HMA-*co*-AAm)	KCl	Conductivity (~0.039 S/cm), stretchability (~1784%), adhesion, self-recovery	[41]
	PEG	Na_2_HPO_4_	Conductivity, stretchability, biocompatibility	[42]
**Crosslinked ion**	DHA/PAAc	Fe(NO_3_)_3_	Conductivity, thermoplastic, adhesion, self-healing	[43]
	KC/PAAm	KCl	GF = 6, stretchability (~950%), antifreeze, self-healing, linear range (0.5–400%)	[44]
	PAAm/Alginate	CaCl_2_	GF = 0.3, stretchability (~1700%), linear range (20%–800%)	[45]
	HPAAm-HLPs/Alginate	NaCl, CaCl_2_	Conductivity, stretchability (~2990%), self-recovery	[46]
	KC/PAAm	KCl	GF = 0.23–0.63, stretchability (~1000%), self-healing, 3D printing, linear range (0–1000%)	[47]
	Agar/Alginate/PAAm	CaCl_2_	GF = 3.83, stretchability (~250%), 3D printing, linear range (0–200%)	[48]
	PVA/CNF	Na_2_B_4_O_7_, CaCl_2_	GF = 0.75, stretchability (~1919%), biocompatible, self-healing	[49]
	PVA	Na_2_B_4_O_7_	GF = 4, stretchability, adhesion, self-healing, linear range (0.1%–500%)	[50]
	Guar gum	Na_2_B_4_O_7_	Conductivity, antifreeze, adhesion	[51]
	PEG-PAMAA	Fe^3+^	Conductivity (0.0016–0.0062 S/cm), stretchability (~1800%), self-healing, linear range (0%–800%)	[52]
	PVA/Alginate/PAAm	CaCl_2_	Conductivity (~1.3 S/cm), stretchability (~959%), self-recovery, linear range (0%–300%)	[53]

**Table 2 materials-13-03947-t002:** Comparison of different types of conductive hydrogel.

Types	Advantages	Disadvantages
**Ionic Conductors**	Low cost, easy preparation, high stretchability, wide linear range (up to ~1600%)	Low conductivity and sensitivity (GF < 10), high salt concentration will damage cells
**Conducting Nanomaterials**	High conductivity and sensitivity (GF_max_ = 7790), stable contact interface	Large deformation will cause the fillers dislocation, limited linear range
**Conducting Polymers**	Uniform dispersion, high conductivity	Few types of conductive polymers

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
