# Peer review of "A Review of Conductive Hydrogel Used in Flexible Strain Sensor"

_materials, 2020, doi:10.3390/ma13183947_

Round 1

Reviewer 1 Report

As attached

Reviewer 2 Report

In the manuscript entitled "A Review of Conductive Hydrogel Used in Flexible Strain Sensor", the authors introduce the synthesis methods of conductive hydrogels and their physical and chemical properties. Also, challenges and future perspectives of the conductive hydrogel-based flexible strain sensor were well discussed.

However, the applications of the flexible strain sensor are also very important. This manuscript reviewed the synthesis methods of conductive hydrogels, but there is no discussion about the application. The authors should add one more figure about the application fields (health monitoring, robot control, or human-machine interface) of the flexible strain sensor using conductive hydrogel and discuss the limits and advantages, it would be helpful to the reader to understand the importance of conductive hydrogel for flexible strain sensor.

Reviewer 3 Report

There has been a recent surge in research on hydrogel-based sensing devices as soft robotics. This review is very timely as their is great interest in flexible sensors. It is also well-organized and written reasonably.

I have three suggestions,

  1. 3D printing of hydrogels to generate complex patterns is highly sought after for preparing wearable devices. The authors should incorporate a few lines about the process.
  2. It is worth stressing the ability of the sensor to repeatedly detect the change in electrical signals under strain (stretching-releasing hysteresis) without significant damage (in the introduction).
  3. In section '3.1 Carbon-based nanomaterials' the authors have highlighted better performance of CNTs and discussed an example of MWCNT crosslinked chitosan hydrogel. There are examples of single-walled CNTs (SWCNTs) in the literature. A brief comparison between the two would be really helpful.
